# acia-workflows: Automated Single-cell Imaging Analysis for Scalable and Deep Learning-based Live-cell Imaging Analysis Workflows

## Abstract

Live-cell imaging technology enables detailed spatio-temporal characterization of living cells at single-cell resolution, which is critical for advancing research across the life sciences, from biomedical applications to bioprocessing. High-throughput setups with tens to hundreds of parallel cell cultivations offer the potential for robust and reproducible insights. However, these insights are hidden within hundreds of GBs of time-lapse imaging data recorded per experiment. Recent advances in state-of-the-art deep learning methods for cell segmentation and tracking now enable the automated analysis of such data volumes, offering unprecedented opportunities to study single-cell dynamics systematically. The next key challenge, however, lies in integrating these powerful tools into accessible, flexible, and user-friendly workflows that support routine application in biological research. In this work, we present `acia-workflows` a platform that combines three key components: (1) the automated live-cell imaging analysis (`acia`) Python library for modular design of image analysis pipelines supporting eight deep learning segmentation and tracking approaches, (2) the design of workflows that assemble the sequential image analysis pipeline, software dependencies, documentation, and visualizations into a single Jupyter Notebook leading to accessible, reproducible and scalable analysis workflows, (3) a collection of application workflows that demonstrate the analysis and customization capabilities in real-world applications. In particular, we present a subset of three application workflows investigating various types of microfluidic live-cell imaging experiments ranging from growth rate comparisons to a precise, minute-resolution quantitative analysis of the response dynamics of individual cells to changing oxygen conditions. Our extensive collection of more than ten application workflows is open source and publicly available at https://github.com/JuBiotech/acia-workflows.

## 1 Introduction

Live-cell imaging is at the forefront of investigating the dynamic behavior of living cells across space and time, fostering our understanding of cancer treatment Alieva et al. [2023], protein secretion Shirasaki et al. [2014], Raphael et al. [2013], diseases Weissmann and Brandt [2008], Campbell and Hope [2008], single-cell heterogeneity Preedy et al. [2024], Huang et al. [2024], and biofilm formation Drescher et al. [2014], Hartmann et al. [2019]. Combining automated live-cell imaging with disposable, high-throughput microfluidic devices (MLCI) enables the simultaneous recording of hundreds of independently developing cell populations within a single experimental run at constant or time-varying conditions, unlocking high-throughput screenings.

Submitted to 39th Conference on Neural Information Processing Systems (NeurIPS 2025). Do not distribute.

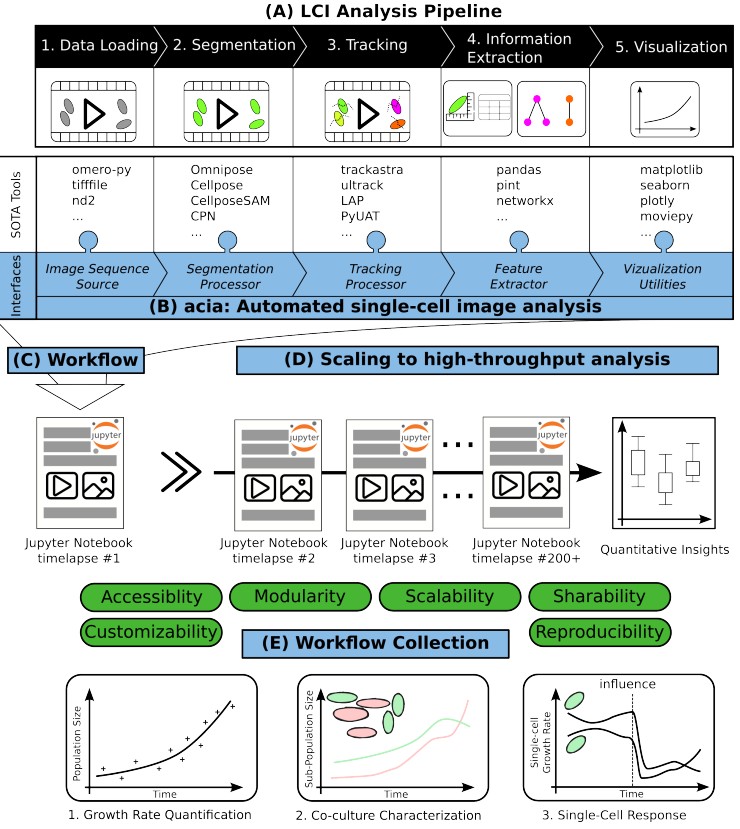

Figure 1: Five-step MLCI analysis pipeline (A), implemented in modular components of the `acia` library (B) utilizing SOTA methods and existing Python libraries. These steps are implemented sequentially within a single Jupyter Notebook, fusing code, documentation, software dependencies, and visualizations into a single workflow (C). This workflow is automatically scaled to high-throughput experiments with numerous time-lapse recordings to gain quantitative insights (D). Our workflow collection (E) showcases the importance of the six key capabilities: *accessibility*, *customizability*, *modularity*, *scalability*, *shareability*, and *reproducibility* (ACMS2R).

The key to the power of MLCI lies in extracting single-cell information from time-lapse data and analyzing the spatio-temporal development of cells and their populations. However, these single-cell measurements are hidden within the imaging data and must be extracted from tens to hundreds of gigabytes of time-lapse data. This extraction requires highly automated image analysis pipelines (Figure 1A).

In recent years, numerous analysis pipelines have been developed that demonstrate the enormous potential of automated image analysis in combination with deep-learning segmentation and tracking O'Connor et al. [2022], Berg et al. [2019], Stirling et al. [2021], Stylianidou et al. [2016], Lo et al. [2024], Ouyang et al. [2019], Luik et al. [2024]. However, these pipelines are usually centered around a specific segmentation or tracking approach, or are plugin-based and require extensive programming to establish a complete image analysis pipeline making them challenging to use in a daily practice and across multi-disciplinary research teams.

## 2 Analysis workflows

Based on our experience of deploying MLCI image analysis pipelines, we have identified the six most important capabilities for such image analysis pipelines: *accessibility*, *customizability*, *modularity*, *scalability*, *shareability*, and *reproducibility* (ACMS2R).

To overcome the limitations of existing pipelines in addressing the six key capabilities, we present `acia-workflows` - a time-lapse analysis platform that integrates SOTA tools into ACMS2R work-

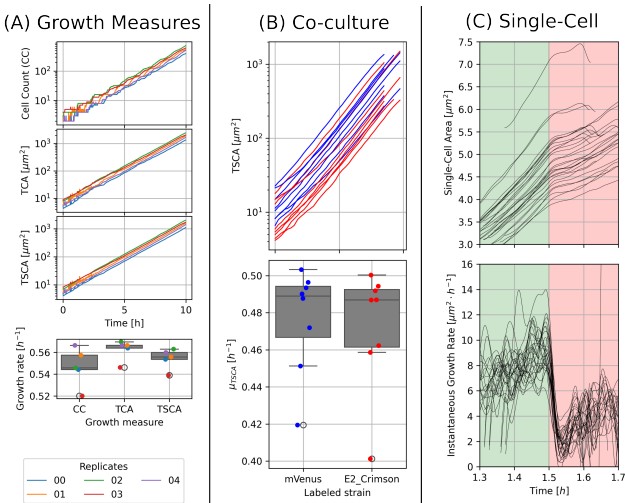

Figure 2: **Scaling MLCI analysis to multiple time-lapses.** (A) shows cell count (CC), total cell area (TCA), and total single-cell area (TSCA) development (top to bottom) for five *C. glutamicum* replicates. The growth rate distribution for all three measures is shown at the bottom. Measurements for the replicates are shown in different colors. (B) shows the *C. glutamicum* co-culture TSCA development (top) and exponential growth rate quantification based on TSCA (bottom) for eight replicates for mVenus (blue) and E2-Crimson (red) labeled strains. (C) shows the single-cell area development (top) and instantaneous growth rates (bottom) of *E. coli* in aerobic (green) and anaerobic phase (red) for five replicates and 30 individual cells.

flows. To achieve this, `acia-workflows` combines three complementary components: First, the `acia` Python library implements the modular time-lapse analysis pipeline (Figure 1B). Second, a workflow concept that enables the integration of code, documentation, and visualizations into a single traceable document (Figure 1C), and third, an open-source collection of application workflows that demonstrates the six key capabilities. In particular, `acia-workflows` deals with image data loading, integrates eight plug-and-play segmentation and tracking methods, extracts important single-cell features such as cell size or heritage, and provides extensive visualization, including videos and interactive lineages. The deployment of these workflows in Jupyter Notebooks facilitates the combination of documentation, coding, and visualization into one traceable and reproducible document and lowers the entry barrier for new users.

## 3 Results

To demonstrate the capabilities of `acia-workflows` and its impact for deep-learning-based single-cell analysis, we present three exemplary case studies: (A) Quantifying population growth rates, (B) performing co-culture characterization, and (C) measuring single-cell responses to changes in cultivation conditions (Figure 2). In these case studies, we present methods for computing cell growth rates from time-lapse videos, demonstrate how to measure and compare growth rates in co-cultures, and measure the speed and homogeneity of adaptation of *E. coli* to alternating oxygen conditions. We leverage the scalability of our workflows to apply them to multiple time-lapse videos, gaining more robust insights without manual code changes. To emphasize accessibility, sharability, and reproducibility, the workflows are available open-source, along with a comprehensive set of over 10 application workflows. These can be reproduced with GPU acceleration directly in the web browser using Google Colab.

Publishing the `acia-workflows` platform with its collection of application workflows, we have lowered the entry barrier of MLCI analysis while showing *customizability* and *reproducibility* for high-throughput analyses. Thus, `acia-workflows` represents an important step toward democratizing image analysis workflows in live-cell imaging, unlocking the potential of single-cell insights for life scientists and data scientists alike.

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

## 4  Potential Negative Societal Impact

This paper provides open-source tools for biomedical and biotechnological research. Thus, potential harm can arise from general misuse of biomedical and biotechnological technologies.

