# OpenReview forum: "acia-workflows: Automated Single-cell Imaging Analysis for Scalable and Deep Learning-based Live-cell Imaging Analysis Workflows"
_EurIPS.cc/2025/Workshop/MedEurIPS — EurIPS 2025 Workshop MedEurIPS Submission_

### Official Review · Reviewer_Zw3W · 2025-10-29
**Nice toolbox, not sure the workshop is a great fit.**

**Rating:** 5
**Confidence:** 4

**Review:**

This paper presents an automated image analysis platform for single-cell live imaging. The platform contains a series of segmentation and tracking approaches, as well as a collection of Jupyter notebooks illustrating their use.

The paper discusses the design choices made for making the library and workflows accessible to users. The results are the jupyter notebooks themselves, and not embedded in the paper.

If accepted, I suppose this could be a nice demo. On the other hand, the toolbox seems to primarily target non-technical users, and it is therefore not clear to me whether the workshop audience is well aligned with the paper. For this reason I will score the paper as borderline reject, but I do not have any reasons to be concerned about its soundness -- only its fit with the workshop. The program chairs should feel very free to opt to include the paper on those grounds.

---

### Official Review · Reviewer_V2qH · 2025-10-31
**Better Suited as a Demo Contribution**

**Rating:** 5
**Confidence:** 4

**Review:**

This work introduces acia-workflows, an open-source platform for automated live-cell imaging analysis. It brings together several state-of-the-art segmentation and tracking methods in a modular, Jupyter-based environment designed for accessibility and reproducibility.

The paper is clearly written and well structured. It addresses an important practical gap in live-cell imaging by making advanced analysis tools easier to use and reproduce. The open-source release and integration into Jupyter workflows make it especially useful for interdisciplinary research teams. However, the contribution is mostly infrastructural and does not introduce new machine learning methods or insights. The evaluation remains qualitative, with no benchmarking or comparisons to related pipelines.

Overall, this is a valuable effort that would make a good demo or community contribution. However, its focus on workflow integration rather than methodological advances makes it less aligned with MedEurIPS’s scope.

---

### Decision · Program_Chairs · 2025-10-31

**Decision:**

Reject

**Comment:**

Both reviewers appreciate the clarity and usefulness of the proposed open-source platform for live-cell imaging analysis. However, they note that the contribution is primarily infrastructural rather than methodological, making it less aligned with MedEurIPS’s focus. The work would be better suited as a demo or community contribution.